# Electrical Resistance Sensing of Epoxy Curing Using an Embedded Carbon Nanotube Yarn

**DOI:** 10.3390/s20113230

**Published:** 2020-06-05

**Authors:** Omar Rodríguez-Uicab, Jandro L. Abot, Francis Avilés

**Affiliations:** 1Department of Mechanical Engineering, The Catholic University of America, Washington, DC 20064, USA; abot@cua.edu; 2Centro de Investigación Científica de Yucatán A.C., Unidad de Materiales, Calle 43 No. 130 x 32 y 34 Col, Chuburna de Hidalgo, 97205 Mérida, Mexico; faviles@cicy.mx

**Keywords:** carbon nanotube yarn, electrical resistance, curing effects, thermosetting matrix, epoxy

## Abstract

Curing effects were investigated by using the electrical response of a single carbon nanotube yarn (CNTY) embedded in an epoxy resin during the polymerization process. Two epoxy resins of different viscosities and curing temperatures were investigated, varying also the concentration of the curing agent. It is shown that the kinetics of resin curing can be followed by using the electrical response of an individual CNTY embedded in the resin. The electrical resistance of an embedded CNTY increased (~9%) after resin curing for an epoxy resin cured at 130 °C with viscosity of ~59 cP at the pouring/curing temperature (“Epon 862”), while it decreased (~ −9%) for a different epoxy cured at 60 °C, whose viscosity is about double at the corresponding curing temperature. Lowering the curing temperature from 60 °C to room temperature caused slower and smoother changes of electrical resistance over time and smaller (positive) residual resistance. Increasing the concentration of the curing agent caused a faster curing kinetics and, consequently, more abrupt changes of electrical resistance over time, with negative residual electrical resistance. Therefore, the resin viscosity and curing kinetics play a paramount role in the CNTY wicking, wetting and resin infiltration processes, which ultimately govern the electrical response of the CNTY immersed into epoxy.

## 1. Introduction

Carbon nanotube yarns (CNTYs) are electro-conductive materials with excellent mechanical, thermal, electrical and piezoresistive properties [1,2,3]. The electrical and thermal conductivity of CNTYs may be similar to that of carbon fibers [4] and their porosity makes them even more attractive for the polymer composites community. The excellent electrical properties of CNTYs make them attractive candidates in various applications such as strain sensing [5,6,7,8,9,10,11,12], supercapacitors, antennas, and other electrical devices [12,13,14]. CNTYs can be integrated into thermosetting polymer composites and used as in situ sensing elements for stress build-up and damage monitoring [3,8,15]. An important concern in thermosetting polymer composites is the development of residual stresses during resin polymerization (curing) [16,17,18,19]. Residual stresses may develop because of polymer shrinkage during curing, and because of the mismatch in thermal shrinkage/expansion between the fiber and matrix [20,21]. The volumetric changes of thermosetting resins during the curing process can be described as a combination of chemical shrinkage at a constant curing temperature, and thermal shrinkage during the cooling curing process [22]. Chemical shrinkage occurs during the generation of the three-dimensional polymer network [19], as a result of the changes from van der Waals bonding to covalent bonding during the macromolecular network formation [23,24]. Thus, it is fairly common that fibers in a cured fiber-reinforced composite exhibit compressive residual stresses [25,26]. These stresses may affect the structure of the fiber and cause fiber waviness, which degrades the mechanical response of the composite [21]. In situ determination of curing stresses has been studied by using the electrical response of carbon fibers [20,25,26,27] or glass fibers [28]. For example, Huang and Young [29], investigated residual stresses in carbon fiber/epoxy composites using Raman spectroscopy, a technique which has proved to be an excellent scientific tool to determine the residual stresses in composites. Their results showed higher residual stresses in specimens cured at higher temperatures, in comparison to specimens cured at room temperature (RT). A more practical alternative to Raman spectroscopy consists of embedding a single carbon fiber into an uncured resin and correlating the change in electrical resistance of the fiber during resin curing with the residual stresses developed. This has been conducted using carbon fibers in the pioneering works of Crasto and Kim [27] and those by Chung et al. [21,25]. Crasto and Kim [27] investigated the effect of the curing temperature of epoxy resin on the residual stresses developed in an individual carbon fiber. Using a single AS4 carbon fiber embedded in an epoxy resin (mixture of Epon 828 and Jeffamine D230), their results showed higher residual stresses for higher curing temperatures, which were correlated with higher increments in electrical resistance. These landmark works proved the concept of monitoring curing stresses in polymers composites using the electrical resistance of carbon fibers. However, the sensitivity of carbon fibers to this aim may be limited by their relatively low surface area (<0.7 m^2^/g [30]), which is in direct contact with the resin. In this regard, the porosity and high surface area of CNTYs (e.g., 70–200 m^2^/g [31,32]) provides an opportunity for increased surface sensitivity, through increased wetting and increased interactions with the viscous resin. The high porosity of CNTYs may also be exploited to promote capillary diffusion of polar liquids, as shown in previous works by Vilatela et al. [31,32]. However, studies on this subject are yet very scarce and several questions remain regarding the electrical response of the CNTY integrated in different thermosetting resins and under different curing conditions. Therefore, this study aims to investigate the curing effects and development of residual stresses during epoxy resin curing, through the electrical response of a single CNTY embedded in the epoxy polymer. The electrical response and temperature of a single CNTY embedded in an epoxy resin were monitored during the curing process of two resins of significantly different viscosities (described in Section 2), and the resulting electrical response and simultaneous temperature profiles are shown in Section 3. To better understand the role of the curing kinetics of the polymer, the curing temperature and stoichiometry ratio of one resin is also investigated.

## 2. Materials and Methods

### 2.1. Materials

The CNTYs were acquired from Nanoworld laboratories of the University of Cincinnati (Cincinnati, OH, USA). The continuous CNTY comprises thousands of twisted multiwall carbon nanotubes (MWCNTs) with a diameter of ~30 µm, a density of ~0.65 g/cm^3^, and a twisting angle of ~30°. CNTYs were spun from forests consisting of carbon nanotubes with a 12 nm outer diameter and a distribution of two to three walls [3]. Two commercial epoxy resins (ER) were used, viz. EPON 862 (labeled “ER-A”, see Table 1) and Toolfusion 1A/1B (labeled “ER-B”, see Table 1), both typically used for polymer composite manufacturing. Epon 862 and its curing agent EPIKURE W were provided by Miller-Stephenson Chemical Co. (Danbury, CT, USA). The mixing ratio of ER-A was 100:23 by weight and was cured at 130 °C for 1.5 h, as recommended by the manufacturer. Toolfusion 1A/1B were provided by Airtech International Inc. (Huntington Beach, CA, USA). For ER-B, three curing conditions were used, as indicated in Table 1. The first one (“ER-B”) employed a stoichiometric ratio of 100:20 (resin:catalyzer) by weight, as recommended by the manufacturer, and was cured at 60 °C for 4 h. To investigate the effect of curing temperature, the second condition employed the same resin-to-curing agent weight ratio (100:20) but was cured at RT (~25 °C) for 10 h. This resin is labeled “ER-B-RT” in Table 1. To investigate the effect of an excess of curing agent, the last condition (“ER-B-50” in Table 1) used a resin-to-curing agent weight ratio of 100:50 (significantly higher than the one recommended by the manufacturer) and was cured at RT for 5 h. Table 1 includes the viscosity (measured) of both epoxy resins used. The viscosity was measured using an ER 2000 rheometer of TA Instruments (New Castle, DE, USA). Further details as well as representative viscosity curves are provided in Appendix A.

The coefficient of thermal expansion (*β*) of the polymers listed in Table 1 was measured by thermo-mechanical analysis (TMA) using a Q400 EM equipment of TA Instruments (New Castle, DE, USA). The samples were heated with an applied force of 0.05 N from 0.25 to 175 °C at 1 °C/min with a ±5 °C amplitude. The samples analyzed by TMA were previously cured in rectangular cuboid shapes of 4 × 4 × 20 mm. To obtain reproducibility, three replicates of each polymer in Table 1 were analyzed. The elastic modulus (*E*) listed in Table 1 was measured using an MTS Criterion 43 universal testing machine (Eden Prairie, MN, USA) with a 30 kN load cell and cross-head speed of 1 mm/min; the corresponding strain was measured using an MTS extensometer model 634.12F-24. The specimens were of type I according to the ASTM standard D638 [33], and eight replicates per group were tested.

### 2.2. Experimental Set Up for In-Situ Electrical Resistance Measurements

Four groups of CNTY/resin (CNTY/ER) specimens with the CNTY placed in a silicon mold and fully embedded in epoxy resin were manufactured as shown in Figure 1a, achieving a final geometry after resin curing as depicted in Figure 1b. Firstly, four parallel 40 AWG copper wires were fixed to the sides of a specially manufactured silicon mold using a needle. These four wires acted as electrodes for four-point probe electrical measurements and were centered at the specimen mid-thickness (Figure 1a). Subsequently, an individual CNTY was placed lying transversely over the copper wires and bonded to the copper wires by using carbon black-based conductive paint (Bare Conductive, London, UK). The use of four-point probe measurement method ensures that the resistance change of the CNTY during the curing process is not affected by the contact resistance between the conductive paint electrodes and the CNTY. The CNTY in all samples was slightly pre-stretched using a constant hanging mass of 116 mg, which corresponds to ~0.1% of the yarn strength for “medium twist” as previously reported [3]. For CNTY/ER-A specimens, the resin and curing agent (*i*) were mixed for 3 min (*ii*) and the mixture was heated to 100 °C for 15 min to eliminate air bubbles (*iii*, ER-A), as indicated in Figure 1a. For CNTY/ER-B specimens, the mixture was placed inside a vacuum chamber to eliminate air bubbles (*iii*, ER-B), and the mixture was poured into the mold containing the CNTY and electrodes. For specimens curing at a temperature of 130 °C (CNTY/ER-A specimens) or 60 °C (CNTY/ER-B specimens), the mold containing the single CNTY and electrodes was pre-heated inside a Fisher Scientific Isotherm 625 G convection oven (Rockville MD, USA) until the oven reached the desired temperature (*iv*), indicated in Table 1. For specimens cured at RT (CNTY/ER-B-RT and CNTY/ER-B-50 specimens), the resin and curing agent were mixed for 3 min, air bubbles were removed using a vacuum chamber, and then the mixture was poured inside the mold and cured for 10 h (CNTY/ER-B-RT specimens) or 5 h (CNTY/ER-B-50 specimens) at RT. As a baseline for comparison, the same procedure was repeated to characterize only the CNTY (freestanding, without resin) in the same mold and identical conditions, but without pouring resin. A three replicate test plan was conducted for all testing.

The electrical measurements were conducted in situ during resin curing, starting few minutes before, and continuing during resin pouring and subsequent curing of the resin poured into the mold. The four-point probe method was used to calculate electrical resistance (*R*), by measuring the voltage drop (*V*) between the inner electrodes, under an applied constant current (*I*), as depicted in Figure 1b. This was performed using a National Instruments (NI) PXI-4072 LCR card mounted in a NI PXI-1033 chassis (Austin, TX, USA). A K-type thermocouple was used for temperature measurement (*T*), placed slightly above the CNTY, as shown in Figure 1b. The thermocouple was connected to an NI 9211 card mounted in a NI cDAQ-9178 chassis. The electrical resistance (*R*) and temperature (*T*) were recorded simultaneously at 1 Hz using the NI Signal Express software.

### 2.3. Curing Program and Data Reduction

CNTY/ER-A and CNTY/ER-B specimens required a temperature program for curing, and the changes in electrical resistance were monitored through the whole curing program. Figure 2 shows the temperature program used for those two groups of resins, divided into four zones. The experiment started by an initial zone of 10 min of stabilization, where the temperature was kept constant at RT (*T*_0_). This region is indicated in Figure 2 as zone I. The electrical resistance at the onset of this zone (very beginning of the experiment) was labeled as *R*_0_. The mold containing the neat CNTY and electrodes was placed inside the oven towards the end of zone I, and subsequently the oven was turned on, rising the temperature from RT to the curing temperature *T*_1_ (130 °C for CNTY/ER-A specimens or 60 °C for CNTY/ER-B specimens), which corresponds to the onset of zone II. In this zone, *R* decreased for CNTY/ER-A and CNTY/ER-B specimens, as will be further discussed. Once the thermocouple over the specimen indicated that *T*_1_ was reached, the door of the oven was opened (for ~2 min) and the resin was poured into the mold containing the CNTY, which indicates the end of zone II and beginning of zone III. The resin then cured at a constant temperature *T*_1_ in zone III (curing zone). The elapsed time in zone III was 1.5 h for CNTY/ER-A specimens (130 °C) and 4 h for CNTY/ER-B specimens (60 °C). The oven was then turned off and the specimen was cooled down, which corresponds to the onset of the last zone (IV). This cooling process took about 5 h for CNTY/ER-A specimens and 3 h for CNTY/ER-B specimens, finally reaching back RT. The electrical resistance increased in zone IV, as will be further discussed. Notice that in Figure 2 there is no “*R*_0_*^I^*”, since it was decided to name *R* corresponding to the onset of zone I as simply *R*_0_.

Table 2 further describes the zones and parameters characterized for CNTY/ER-A and CNTY/ER-B specimens.

At zone I (no resin added, only freestanding CNTY is present in the mold), small oscillations of *ΔR*/*R*_0_ exist under a constant *T*, so the signal-to-noise ratio of *R* for the CNTY (*SNR_CNTY_*) was calculated according to:*SNR_CNTY_* = 10 *Log*_10_ (*ΔR*/*R*_0_*^Mean^*/*ΔR*/*R*_0_*^SD^*)(1)
where *ΔR*/*R*_0_*^Mean^* is the mean value of *ΔR*/*R*_0_ in those 10 min, and *ΔR*/*R*_0_*^SD^* is the corresponding standard deviation.

At zone II, there is only a dry freestanding CNTY during the heating process, and a temperature coefficient of resistance (*α_H_^CNTY^*) was defined as the slope of the approximately straight line in such a zone, i.e.,:*α_H_^CNTY^* = (*ΔR*/*R*_0_)*_II_*/*ΔT_II_*(2)
where (*ΔR*/*R*_0_)*_II_* = (*R*_0_*^III^* − *R*_0_*^I^**^I^*)/*R*_0_*^II^* and *ΔT_II_* = *T*_1_ − *T*_0_.

Zone III is a zone of constant temperature, and only small random fluctuations in *R* were detected. However, resin pouring occurs at the onset of this zone, and there was a marked change in *R* due to resin pouring. To quantify this effect, a parameter named *ERP* (effect of resin pouring) was calculated to capture the difference between (*ΔR*/*R*_0_)*_III_* at the onset of zone III for CNTY/epoxy resin specimens and the corresponding (*ΔR/R_0_*)*_III_^CNTY^* at the onset of zone III for the freestanding CNTY, such as,
*ERP* = (*ΔR*/*R*_0_)*_III_* − (*ΔR*/*R*_0_)*_III_^CNTY^*(3)

Finally, zone IV is a cooling zone for the specimen containing the CNTY and (partially cured) resin, so a thermoresistive sensitivity for the CNTY embedded in the resin during the cooling process (*α_C_*) was calculated from this zone as,
*α_C_* = (*ΔR*/*R*_0_)*_IV_*/*ΔT_IV_*(4)
where (*ΔR*/*R*_0_)*_IV_* = (*R*_f_ − *R*_0_*^IV^*)/*R*_0_*^IV^* and *ΔT_IV_* = *T*_1_ − *T*_0_.

Similarly, at zone IV, for the specimen containing the freestanding CNTY (without resin), a temperature coefficient of resistance (thermoresistive sensitivity) during its cooling (*α_C_^CNTY^*) was calculated also using Equation (4) but with (*ΔR*/*R*_0_)*_IV_* corresponding to that of the specimen without resin (just the CNTY).

Finally, it is of interest to calculate the change in electrical resistance of the CNTY before and after the whole curing experiment, i.e., the residual change in electrical resistance (*ΔR*/*R*_0_)*_RES_*, which was quantified as,
(*Δ**R*/*R*_0_)*_RES_* = (*R*_f_ − *R*_0_)/*R*_0_(5)
where *R*_0_ is *R* at the beginning of zone I (at the beginning of the experiment) and *R*_f_ is *R* at the end of zone IV (after polymer curing), as indicated in Figure 2.

CNTY/ER-B-RT and CNTY/ER-B-50 specimens cured at RT, so they did not use a temperature program. For the experiments of these two groups of specimens, *T* and *R* were first stabilized for 10 min and then the resin was poured into the mold containing the CNTY and electrodes, as depicted in Figure 1. The resin was then covered with a Petri dish and left to cure for 10 h at RT for CNTY/ER-B-RT specimens, and for 5 h for CNTY/ER-B-50 specimens. Only the parameters stated in Equations (1) and (5) were calculated for those two groups of specimens.

Additionally, the fracture surface of solid CNTY/ER coupons deliberately broken using tensile tests were examined by scanning electron microscopy (SEM). SEM was conducted by a SU-70 Hitachi microscope operated at 20–25 kV. The fracture surface of the coupons was covered with a thin layer of sputtered gold (15–20 nm).

## 3. Results

### 3.1. Electrical Response of CNTY During Resin Curing Using a Temperature Curing Program

Figure 3a shows measurements of the fractional electrical resistance change (*ΔR*/*R*_0_) and temperature change (*ΔT*) against curing elapsed time for a representative CNT/ER-A specimen, using the temperature program described in Figure 2. In order to better assess the effect of resin pouring and wetting of the CNTY, Figure 3b reports an identical temperature program to that plotted in Figure 3a, but where the resin was not poured into the silicon mold (only a single, dry, freestanding CNTY was present in the mold for the full duration of the experiment). Thus, the electrical response plotted in Figure 3b is purely due to the thermoresistive response of the CNTY, and comparison of Figure 3a,b should allow examining the effect of the resin wetting the CNTY.

A close-up of each of the four zones depicted in Figure 2 and Table 2 is presented in Figure 4. In Figure 3a and Figure 4a, it is observed that during the first 10 min of stabilization (zone I) the temperature of the CNTY was kept fairly constant, with only small oscillations in the electrical resistance. These small oscillations (see numerical scales in both vertical axes) are believed to be due to electronic noise [34,35,36] and dielectric relaxation effects [37].

To evaluate the level of noise in the electrical measurements, the signal to noise ratio (SNR) was calculated according to Equation (1) and is listed in the first row of Table 3. Zone II corresponds to the heating ramp of CNTY (without resin) from RT (~25 °C) to the curing temperature of the resin (130 °C), Figure 4b, where the single CNTY is held still within the mold. Since the CNTY is freestanding in this case, this means that a (heating, *H*) temperature coefficient of resistance (*α_H_^CNTY^*) can be extracted for the CNTY from this zone, as shown in the inset of Figure 4b and listed in the second row of Table 3. The coefficient of determination (*r*^2^) of the linear fit is also indicated in the inset. At zone II, the temperature increases from RT to curing temperature and *ΔR*/*R*_0_ decreases (negative value of *ΔR*/*R*_0_), which is attributed to the intrinsic thermoresistive behavior of the CNTY. The door of the oven was opened and resin was poured into the mold at the end of zone II (onset of zone III), and zone III corresponds to a dwell time (curing process) where *ΔT* is held constant for 1.5 h, Figure 4c. The effect of resin curing on the electrical resistance was calculated according toEquation (3), and is listed in the fifth row of Table 3 (zone III). Figure 3a and Figure 4c show a sharp increase of *ΔR*/*R*_0_ from ~ −7.5% to 3.9% at the beginning of zone III (right after resin pouring). After that, *ΔR*/*R*_0_ decreases from ~3.9% to −3%, and tends to stabilize towards the end of zone III. This behavior could be attributed to initial wicking and infiltration of the CNTY (sudden positive *ΔR*/*R*_0_), followed by the initiation of chemical shrinkage of the polymer matrix [22], which may cause the subsequent decrease in *ΔR*/*R*_0_ (negative *ΔR*/*R*_0_). Electron donor transfer from the resin to the CNTY is also a possibility that had been previously reported as the mechanism which decreases *ΔR*/*R*_0_ [38] (negative *ΔR*/*R*_0_, i.e., increased conductivity). Additional analysis of a separate experiment involving only a CNTY (without resin) under the same conditions is provided in Appendix A. Zone IV corresponds to the cooling process of the CNTY and resin from the curing temperature (130 °C) to RT (~25 °C), after the oven is turned off. Figure 4d shows that in zone IV, *ΔR*/*R*_0_ largely increases (from ~ −3% to 11%) while the temperature decreases. Part of this electrical response is undoubtedly due to the intrinsic (negative) thermoresistivity of the yarn, but other effects must be contributing. Thermal polymer shrinkage is expected to occur during this zone [19,22,39], concomitant with the development of residual stresses in the CNTY, and they may be factors contributing to the (large) increase of *ΔR*/*R*_0_ in zone IV. At zone IV, the thermoresistive sensitivity of the CNTY alone (*α_C_^CNTY^*) and that of the CNTY embedded in the resin (*α_C_*) was calculated according to the inset of Figure 4d and Equation (4), and both are also listed in Table 3. The fact that *α_C_* and *α_C_^CNTY^* are different points to physicochemical interactions between the resin and CNTY, due to a combination of various mechanisms, will be further discussed. At the end of the curing cycle in Figure 3, it is seen from Figure 3a that (*ΔR*/*R*_0_)*_RES_* reaches ~11%, but the same experiment using only a freestanding CNTY without pouring resin (Figure 3b) reaches (*ΔR*/*R*_0_)*_RES_* ~ −1%. That is, after finalizing the experiment, *ΔR*/*R*_0_ for the freestanding CNTY (Figure 3b) almost comes back to the initial zero value of *ΔR*/*R*_0_, but specimens with resin ER-A (CNTY/ER-A specimens) showed a very different behavior, with positive residual (*ΔR*/*R*_0_)*_RES_*, see Table 3. Such a difference can be attributed to the physical and chemical interactions between the polymer and the CNTY, i.e., wetting, wicking, resin infiltration, electron donor transfer, chemical and thermal shrinkage, and the development of residual stresses within the porous yarn.

Figure 5 shows the fractional change of the electrical resistance (*ΔR*/*R*_0_) and temperature change (*ΔT*) of a representative CNTY/ER-B specimen (Figure 5a) and freestanding CNTY (Figure 5b), using the temperature program depicted in Figure 2 with T_1_ = 60 °C.

Details of each zone of the curing process of CNTY/ER-B are shown in Figure 6. At zone I (not shown) and zone II (Figure 6a), the values of *SNR^CNTY^* and *α_H_^CNTY^* of CNTY/ER-B specimens (Figure 5a) were similar to those of CNTY/ER-A specimens (see Table 3), since actually only the freestanding CNTY is present in both experiments. However, *ΔR*/*R*_0_ for CNTY/ER-B specimens at zone III decreases from ~ −1.9% (at the beginning of zone III) to ~ −11.7% (at the end of the zone III, Figure 6b). This behavior could be attributed to the onset of polymerization and chemical shrinkage of the polymer ER-B [19,22], and probably affected by electron transfer from the resin [38]. For the freestanding CNTYs, the value of *ΔR*/*R*_0_ does not show significant changes at zone III (see Appendix A) in comparison to CNTY/ER-B. However, the individual CNTY shows sensitivity in *ΔR*/*R*_0_ (~2%) when the door of the oven was opened. At zone IV, the value of (*ΔR*/*R*_0_)*_RES_* at the end of the experiment was ~ −9.5% for CNTY/ER-B (Figure 5a and Figure 6c) and ~ −9.1% for the CNTY itself (Figure 5b).

It is expected that the polymer infiltration effect on (*ΔR*/*R*_0_)*_RES_* is smaller for CNTY/ER-B than for CNTY/ER-A, due to its higher viscosity at the pouring temperature (see Table 1), which inhibits resin infiltration into the porous yarn structure. Since CNTY/ER-B specimens cured at (and were heated to, Figure 5) only 60 °C, it is expected that polymer chemical shrinkage (zone III) [19,22] contributes less than thermal shrinkage and development of thermal stresses (zone IV) to (*ΔR*/*R*_0_)*_RES_*. Notice from Figure 5b and the inset of Figure 6c that the intrinsic thermoresistivity of the CNTY is a strong contributor to the electrical response of CNTY/ER-B specimens. As a reference, Crasto and Kim [27] also observed a negative coefficient of resistance for AS4 carbon fibers, and an increased electrical resistance with epoxy resin curing at RT. A summary of the measured parameters is listed in Table 3.

According to Table 3, at zone I, the average SNR of the CNTY was 29 dB, corresponding to a variation of ~0.03% of the average *ΔR*/*R*_0_ during the initial stabilization process of *R*. At zone II, *α_H_^CNTY^* was measured as −8.5 × 10^−4^ °C^−1^. A negative value of *α_H_^CNTY^* had also been previously documented for freestanding CNTYs [2,40], individual carbon nanotubes [41] and carbon fibers [42]. However, for macroscopic assemblies of axially aligned CNT fibers directly spun from floating catalyst chemical vapor deposition furnace, Lekawa-Raus et al. [43] observed a non-monotonic and nonlinear resistance behavior for temperatures between 68 K and 273 K. They also suggested that their longitudinal electrical resistance is limited by the morphology and composition of the CNTs, rather than by contact resistance between CNTs.

Notice that the specimens with higher curing temperature (CNTY/ER-A, 130 °C) showed higher electrical changes, which are attributed to higher thermal shrinkage and development of higher axial and radial compressive residual stresses [27]. It is known that the mismatch between the coefficients of thermal expansion of the CNTY and epoxy resin leads to generation of residual thermal stresses during the epoxy resin curing and subsequent cooling, especially if they are cured at high temperature [16,27]. For a simplified analysis, assuming a unidirectional state of stress, a solid cross-section area, and perfect CNTY/polymer interphase, the residual stress in the axial direction of the CNTY (*σ_f_*) after polymer curing can be estimated according to [25,26]:(6)$$\sigma_{f}=\frac{E_{f}E_{m}V_{m}\left(\beta_{m}-\beta_{f}\right)\Delta T}{\left(V_{m}E_{m}+V_{f}E_{f}\right)}$$
where *σ_f_* is the longitudinal residual stress built up in the fiber, subscripts *m* and *f* refer to the matrix (epoxy resin) and fiber (CNTY), respectively, *E* is the elastic modulus, *V* is the volume fraction, *β* is the coefficient of thermal expansion and *ΔT* is the temperature difference (between curing temperature and RT). For this material system, *E_f_* = 50 GPa [3], the elastic moduli of the epoxy matrices are *E_m_* = 3.06 GPa (ER-A, Table 1) and *E_m_* = 3.26 GPa, (ER-B, Table 1), which were measured herein using dog-bone specimens, *V_m_* = 0.99997 (≈1), *V_f_* = 3.3 × 10^−5^, *β_f_* = −1.6 × 10^−6^ K^−1^ [1], *β_m_* = 179.5 × 10^−6^ K^−1^ (ER-A, Table 1), and *β_m_* = 56.4 × 10^−6^ K^−1^ (ER-B, Table 1), which were measured herein using a TMA equipment. Using Equation (6), *ΔT* = 105 °C for CNTY/ER-A specimens and *ΔT* = 35 °C for CNTY/ER-B specimens, the residual axial stress in the fiber yields 3.42 GPa for CNTY/ER-A and 0.84 GPa for CNTY/ER-B. The higher electrical changes and (*ΔR*/*R*_0_)*_RES_* for CNTY/ER-A specimens are in agreement with the largest residual stresses developed during curing, induced by the higher volumetric shrinkage of this thermosetting polymer cured at higher temperature [16,27]. However, the simple unidimensional model of Equation (6) does not consider several other relevant factors such as the viscosity of the polymer, the porosity of the CNTY, the polymer infiltration effect, chemical shrinkage, and development of compressive radial stresses during the curing process.

It is worth noticing from Figure 3a and Figure 5a that after the resin cured, the electrical resistance of the CNTY did not return to its initial value. In fact, according to Table 3, (*ΔR*/*R*_0_)*_RES_* was positive for CNTY/ER-A specimens and negative for CNTY/ER-B specimens, see Figure 7. This suggests that the polymer properties play a paramount role in the electrical response of the CNTY soaked in resin. Both values are plotted in Figure 7 along with the corresponding value when the CNTY alone (no resin) is left to experience the same temperature program as ER-A (up to 130 °C) and ER-B (up to 60 °C). The positive value of (*ΔR*/*R*_0_)*_RES_* for CNTY/ER-A specimens (average of 9.0%) indicates that *R*_f_ > *R*_0_, and the fact that (*ΔR*/*R*_0_)*_RES_* of the freestanding CNTY (left side of Figure 7) corresponding to the same temperature program is negative further indicates that the electrical response of CNTY/ER-A specimens is strongly influenced by thermal shrinkage and residual stresses of the epoxy resin, and not only by the instinct thermoresistivity of the CNTY. On the other hand, the residual electrical resistance of CNTY/ER-B specimens (average of −9.0%) is negative and similar to that experienced by its corresponding freestanding CNTY (right side of Figure 7). This fact suggests that, for this resin, which is more viscous at the moment of pouring and cured at only 60 °C, resin infiltration and build-up of residual stress may not represent large contributions, so the yarn thermoresistivity and resin chemical shrinkage may dominate. Indeed, it has been pointed out that when a CNTY becomes in contact with an epoxy resin, the resin wicks and infiltrates the gaps between the bundles of the CNTY [1,44,45], which is expected to cause an increase in *R*. The negative value of (*ΔR*/*R*_0_)*_RES_* for CNTY/ER-B specimens is thus ascribed to its lower curing temperature (60 °C) and higher curing viscosity, which yields lower infiltration and lower curing effects than that of CNTY/ER-A specimens (cured at 130 °C). It should be kept in mind that as the temperature increases, the viscosity of the resin greatly decreases (as indicated by the viscosity listed in Table 1 and the rheometry measurements in Appendix A), which is expected to promote further resin infiltration.

### 3.2. Electrical Response of CNTY during Resin Curing at Room Temperature

The last two material combinations listed in Table 1 (CNYT/ER-B-RT and CNTY/ER-B-50 specimens) were cured at RT, and their electrical response during resin curing is shown in Figure 8. The curing process of CNTY/ER-B-RT specimens was about 10 h (Figure 8a) and about 5 h for CNTY/ER-B-50 specimens (Figure 8b). The experiments were conducted at RT and covered by a Petri dish. As seen from both figures, only small fluctuations in temperature (~1 °C for CNTY/ER-B-RT and ~0.5 °C for CNTY/ER-B-50, solid red triangles) were registered during the experiments, indicating negligible curing exotherms and negligible influence of CNTY thermoresistivity during the curing process (see Appendix A). Thus, the electrical behavior of the CNTY could be only attributed to the interaction between the CNTY and the resin during the curing process, and any potential Joule effect by the small current flow could be discarded. A SNR = 28.4 (±1.8) dB was calculated for the first 10 min of the experiments, which coincides with the values reported in Table 3 extracted from the experiments associated with CNTY/ER-A and CNTY/ER-B. After that, the resin was poured in the mold containing the CNTY.

An instantaneous sharp increase in *ΔR*/*R*_0_ is observed upon resin pouring, which could be ascribed to resin wicking and sudden infiltration into the first (surface) layers of the CNTY. For CNTY/ER-B-RT specimens, the initial sharp increase of *ΔR*/*R*_0_ is ~7.8% (Figure 8a), and ~7.2% for CNTY/ER-B-50 (Figure 8b). This rapid polymer infiltration is expected to be triggered by capillary forces arising from the large fiber porosity [38]. Elastocapillary effects upon resin infiltration yielding small CNTY swelling is also a possibility [32]. After this point, *ΔR*/*R*_0_ decreases to ~3.8% for CNTY/ER-B-RT, and far more, up to ~ −5.1% for CNTY/ER-B-50 specimens. This reduction in electrical resistance may be due chemical shrinkage (polymerization onset) as well as electrochemical interactions between the CNTY and amines of the curing agent, which are electron donors and may act as dopants. Similar decreases of electrical resistance have been observed upon soaking CNTYs in propylamine, an electron donor [38]. Chemical shrinkage during resin cross-linking of CNTY/ER-B-50 specimens may also contribute to the decrease in *R* observed at this point, triggering compressive radial stresses which decrease the CNTY porosity [25,39]. The larger negative value of *ΔR*/*R*_0_ for CNTY/ER-B-50 after resin pouring with respect to CNTY/ER-B-RT is attributed to the faster polymerization kinetics (larger chemical shrinkage) for CNTY/ER-B-50 specimens, which have a deliberate excess of curing agent. The (deliberate) excess of curing agent (50:50) for CNTY/ER-B-50 specimens also provides further concentration of amines, which may act as electron donor molecules for the CNTY [38]. After the *ΔR*/*R*_0_ drop, an inflection point (change in concavity of the curve) is detected in the *R* curve, which approximately coincides with the gel point. The gel point of both epoxy resins is indicated as a vertical dashed line in the plots, and was calculated following the procedure described in the ASTM standard D2471 [46]. The gel point occurred at 143 (±20) min for CNTY/ER-B-RT (Figure 8a) and 66 (±3) min for CNTY/ER-B-50 (Figure 8b). At this point, *ΔR*/*R*_0_ shows a change in the slope sign in the curve, which is attributed to the cross-linking and formation of the resin three-dimensional network [23,24]. This transition is of course significantly more marked for CNTY/ER-B-50 specimens, which have an excess of curing agent and thus experience faster polymerization kinetics. After this point, the polymerization reaction and cross-linking is largely developed, and the resin experiences chemical shrinkage at constant curing temperature [19]. After polymerization, (*ΔR*/*R*_0_)*_RES_* = 4.1% for CNTY/ER-B-RT specimens and (*ΔR*/*R*_0_)*_RES_* = −1.7% for CNTY/ER-B-50 specimens, Figure 8c. The more abrupt changes in *ΔR*/*R*_0_ during polymerization and the larger (*ΔR*/*R*_0_)*_RES_* for CNTY/ER-B-50 specimens are attributed to their non-stoichiometric ratio (excess of curing agent), which yields faster curing kinetics, higher chemical shrinkage and more amine molecules available for electrochemical interactions.

### 3.3. Electron Microscopy Observations of CNTY Wetting

Figure 9 shows representative SEM images of an individual CNTY and the (tensile) fracture surface of CNTY/solid polymer at 1000× (image above) and 5000× (image below). As seen from these images, individual CNTYs (Figure 9a) show a surface texture with high porosity and twisted bundles of carbon nanotubes, characteristic of CNTYs [1]. The gaps between bundles offer large surface area for potential infiltration of the resin, and it has been pointed out that this is more likely to occur inter-bundles in opposition to intra-bundles [1,44]. Figure 9b shows the fractured surface morphology of a solid CNTY/ER-A specimen, with the lower image focusing on the specimen interface. It can be observed that the CNTY is unbroken, no fiber/matrix debonding is detected, and matrix cracking is observed, indicating a strong fiber/matrix adhesion. The external surface of the yarn appears densified, suggesting inter-bundle resin infiltration. This infiltration is attributed to the low viscosity of the ER-A polymer at the curing temperature of 130 °C (Table 1), assisted by the volumetric chemical and thermal shrinkage during the curing process. Figure 9c shows the fracture surface morphology of a CNTY/ER-B specimen, with only modest evidences of yarn surface infiltration. Close to the lower-right (~45°) edge of the interface in the higher magnification image, a zone of fiber/matrix debonding is captured, which may be indicative of a weaker interface. This weaker interface (with respect to CNTY/ER-A) could be triggered by less resin infiltration, since ER-B is more viscous at the curing temperature and cures at a lower temperature. The situation is similar for CNTY/ER-B-RT specimens (Figure 9d), with modest evidences of resin infiltration. For CNTY/ER-B-50 specimens (Figure 9e), the CNTY is seen protruding and pulling out from the ER-B-50 polymer, leaving a relatively smooth surface on the resin. This comparatively weak interface for CNTY/ER-B-50 specimens could be attributed to the non-stoichiometric ratio (high concentration of curing agent) and fast curing kinetics of this resin, and consequently shorter time for resin infiltration [45].

## 4. Conclusions

The in situ electrical resistance response of an individual carbon nanotube yarn (CNTY) embedded in an epoxy resin during polymerization was investigated for four groups of epoxy resins. Two resins required a temperature program, either using high (CNTY/ER-A, 130 °C) or low (CNTY/ER-B, 60 °C) curing temperatures, while the other two were cured at room temperature, either at its corresponding stoichiometry ratio (CNTY/ER-B-RT) or with an excess of curing agent (CNTY/ER-B-50). In order to separate the effects, the thermoresistive response of individual freestanding CNTYs (without epoxy resin) was determined using the same temperature program and conditions as for the resin specimens. The results indicate that several mechanisms concurrently influence the electrical response of the yarn embedded in an epoxy resin during curing, viz. wicking, ingress of resin in the porous structure of the CNTY, electrochemical charge transfer, chemical and thermal shrinkage, and the development of residual stresses. For epoxy resins employing a temperature curing program, a quick increase in electrical resistance (*R*) is seen upon resin pouring, which is associated with capillary effects such as wicking and ingress of resin to the external layers of the porous yarn. This peak in *R* is followed by a more gradual decrease in *R* at constant temperature during resin cross-linking, associated with chemical shrinkage and electron-donor transfer mechanisms. Subsequently, during cooling and resin solidifying, residual stresses build up, which tend to increase *R*. The residual *R* once the (solid) resin has been cured depends on the curing temperature and resin viscosity/properties. The resin with the lower viscosity (at the curing temperature) and the higher curing temperature (CNTY/ER-A) shows positive residual changes of electrical resistance (+9%), while the one with the lower curing temperature (CNTY/ER-B specimens) and higher viscosity shows negative values (−9%).

The specimens which cured at room temperature show a markedly different behavior. After the initial peak (sharp increase) of *R* concomitant with resin pouring (wicking), the electrical resistance decreased which corresponded to the polymerization onset (resin cross-linking). The slower curing kinetics of CNTY/ER-B-RT specimens yielded fewer residual stresses, less changes in *R* during curing, and positive residual *ΔR* after polymerization. On the other hand, the nonstoichiometric ratio (deliberate excess of curing agent) of CNTY/ER-B-50 specimens yielded a fast curing kinetics, concomitant with more marked transitions in *R* and small negative residual values of *ΔR* after polymerization. This is attributed to a higher contribution of resin chemical shrinkage and electron donor transfer from the excess of amines. In both resins curing at room temperature, the gel point was identified from the electrical *R* curve, even when no exotherm was detected during the cross-linking reaction. Thus, wicking, resin infiltration, reduction of porosity, change in the bundle-to-bundle contact resistance, chemical and thermal shrinkage, electron transfer mechanisms with the resin and/or curing agent, as well as development of (radial and longitudinal) residual stresses in the CNTY are believed to be the major contributors on the changes in the electrical resistance experienced by a CNTY immersed in epoxy resin during resin polymerization. Knowledge of the electrical response of the yarn during resin curing, along with the excellent mechanical properties of the yarn, may boost the development of multifunctional intelligent materials integrated into structural composites, which could self-sense exothermic reactions, pinpoint gel times, and track resin curing kinetics.

## Figures and Tables

**Figure 1 sensors-20-03230-f001:**
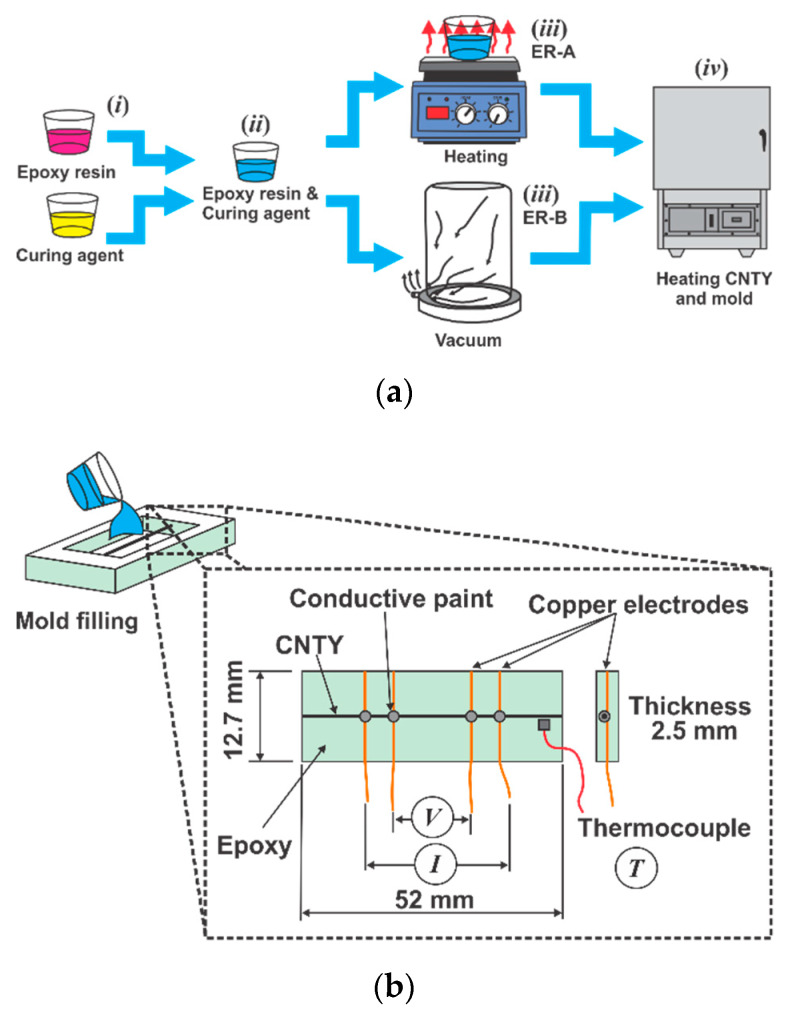
Manufacturing and in situ electrical measurements of carbon nanotube yarns (CNTY)/epoxy specimens. (**a**) Sequential steps for specimen manufacturing (mold filling), (**b**) in-situ electrical measurements and dimensions of CNTY/epoxy specimens. Steps related to heating are not used for epoxy resin (ER)-B-RT and ER-B-50.

**Figure 2 sensors-20-03230-f002:**
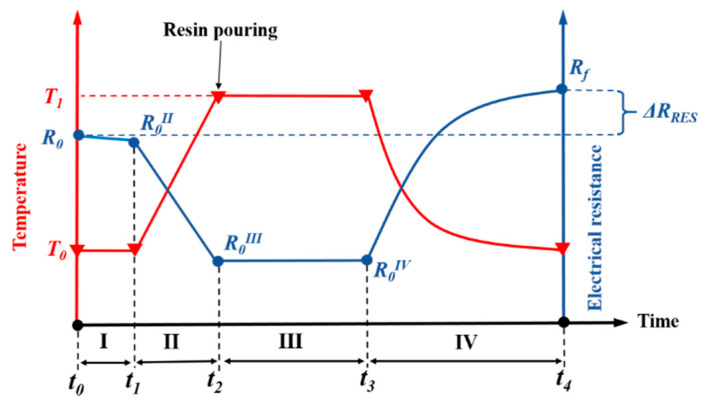
Curing temperature program for CNTY/ER-A and CNTY/ER-B specimens and sketched electrical resistance variation, divided into four zones.

**Figure 3 sensors-20-03230-f003:**
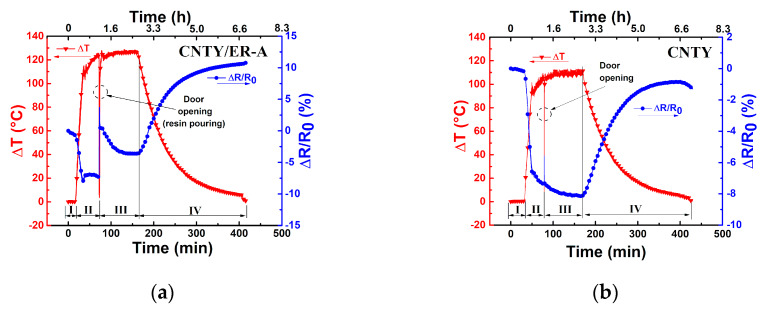
Change in electrical resistance and temperature during the curing resin process of CNTY/ER-A. (**a**) CNTY/ER-A, (**b**) freestanding CNTY under the same temperature program.

**Figure 4 sensors-20-03230-f004:**
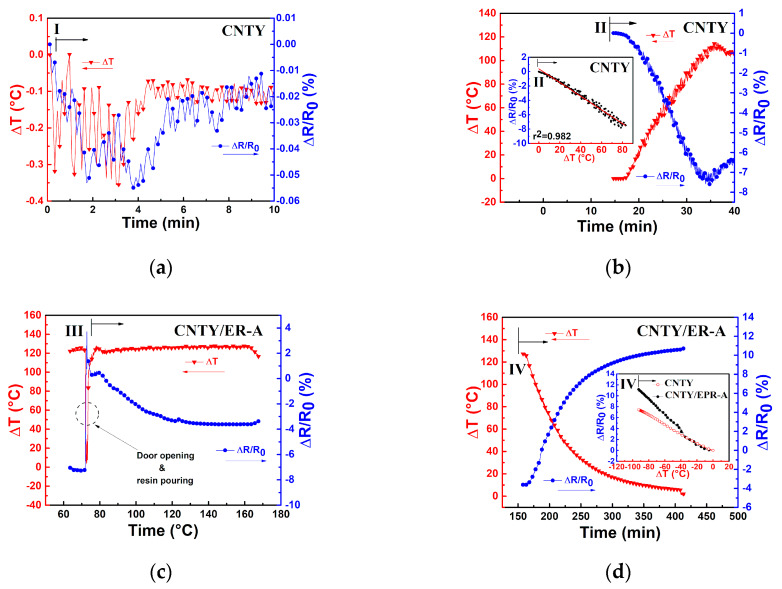
Close-ups of the zones shown in Figure 3 for CNTY/ER-A. (**a**) Zone I (freestanding CNTY), (**b**) heating thermoresistive sensitivity in zone II (freestanding CNTY), (**c**) onset of zone III indicating the moment of resin pouring (CNTY embedded in resin), (**d**) thermoresistive sensitivity in zone IV (CNTY embedded in resin).

**Figure 5 sensors-20-03230-f005:**
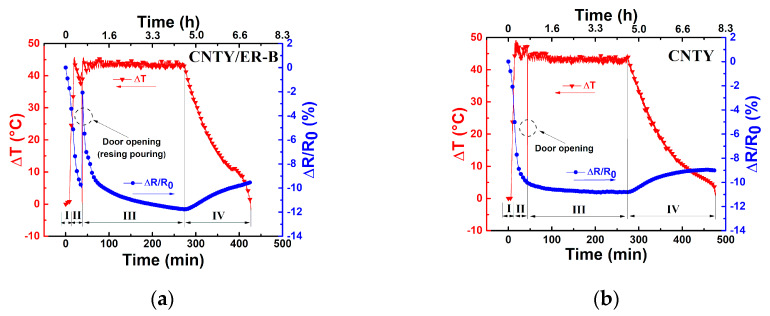
Change in electrical resistance and temperature during the curing process of ER-B. (**a**) CNTY/ER-B, (**b**) freestanding CNTY under the same temperature program.

**Figure 6 sensors-20-03230-f006:**
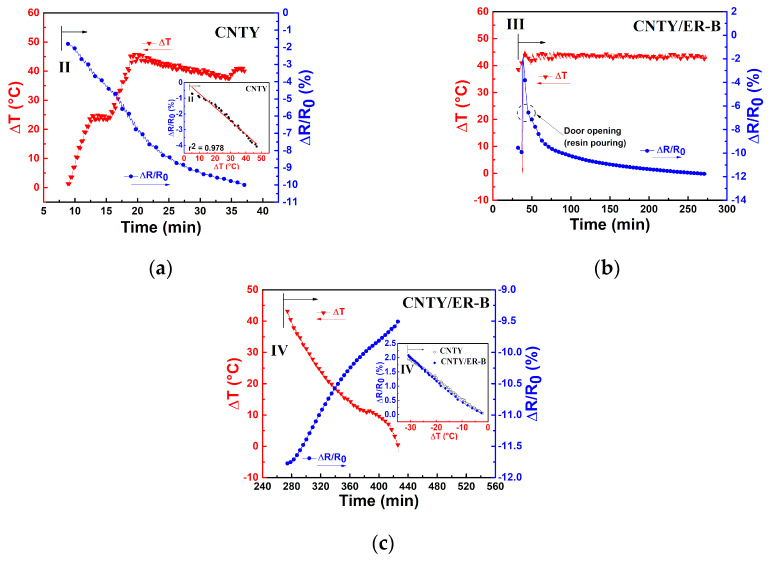
Close-ups of the zones shown in Figure 5 for CNTY/ER-B. (**a**) Heating thermoresistive sensitivity in zone II (freestanding CNTY), (**b**) zone III indicating the moment of resin pouring, (**c**) cooling thermoresistive sensitivity in zone IV (CNTY embedded in resin).

**Figure 7 sensors-20-03230-f007:**
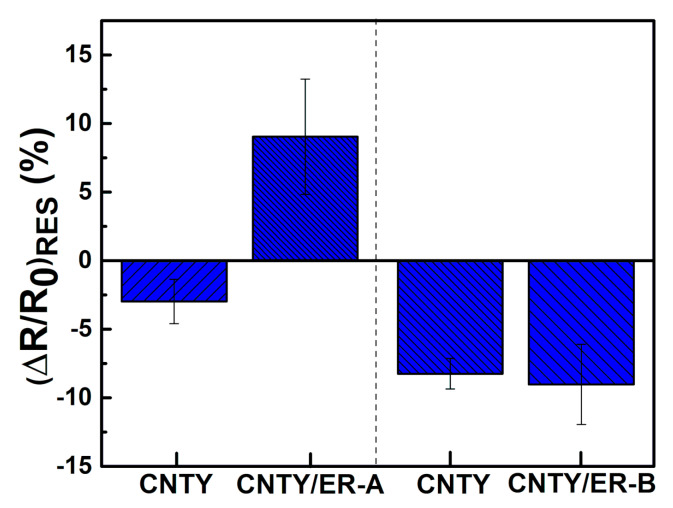
Normalized residual electrical resistance of CNTY/ER-A, CNTY/ER-B and their corresponding freestanding CNTYs under the same temperature program.

**Figure 8 sensors-20-03230-f008:**
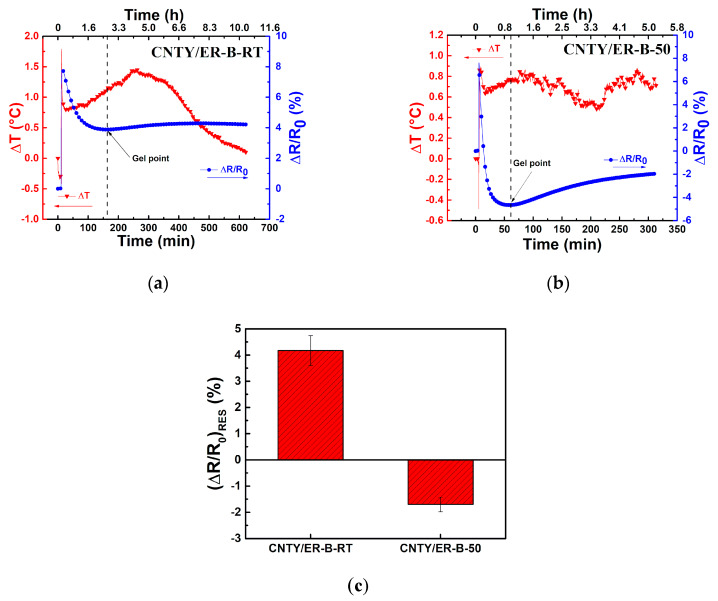
Change in electrical resistance and temperature during the curing process of CNTY/ER-B-RT and CNTY/ER-B-50, both at RT. (**a**) CNTY/ER-B-RT, (**b**) CNTY/ER-B-50, (**c**) normalized residual electrical resistance of CNTY/ER-B-50 and CNTY/ER-B-RT.

**Figure 9 sensors-20-03230-f009:**
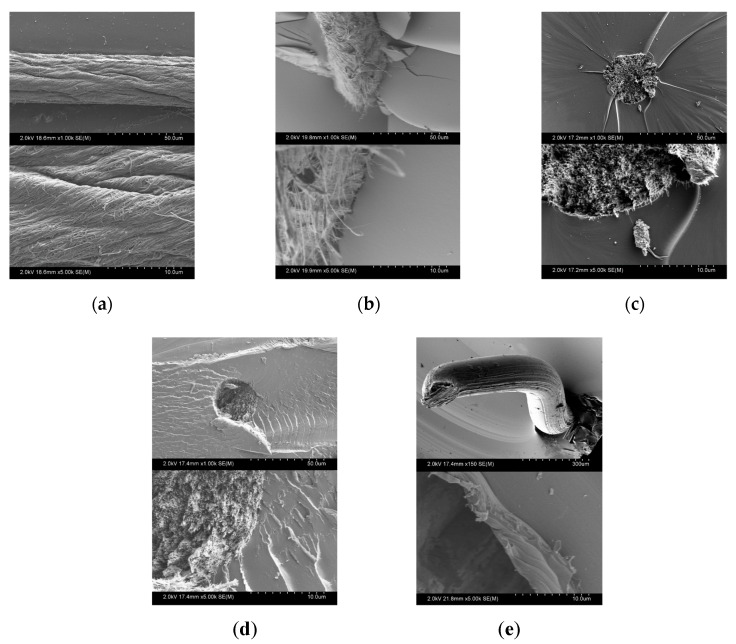
SEM images (1000×, above and 500×, below) of CNTY and CNTY/polymer specimens. (**a**) Individual CNTY, (**b**) CNTY/ER-A, (**c**) CNTY/ER-B, (**d**) CNTY/ER-B-RT, (**e**) CNTY/ER-B-50.

**Table 1 sensors-20-03230-t001:** Nomenclature and physical properties of epoxy resins.

Label	Description and Curing Conditions	Physical Property		
		Liquid Viscosity, (cP) *	Coefficient of Thermal Expansion, *β* (×10^−6^ K^−1^)	Elastic Modulus, *E* (GPa)
ER-A	EPON 862 and EPIKURE W 100:23 by weight cured @ 130 °C for 1.5 h	1625 ± 8	57.8 ± 0.2	3.06 ± 0.16
		@ 30 °C	(0.25 to 25 °C)	
		59 ± 19	179 ± 0.6	
		@ 130 °C	(100 to 175 °C)	
ER-B	Toolfusion 1A/1B 100:20 by weight cured @ 60 °C for 4 h		56.4 ± 0.8	3.26 ± 0.16
			(0.25 to 25 °C)	
			170 ± 4.2	
			(100 to 175 °C)	
ER-B-RT	Toolfusion 1A/1B 100:20 by weight cured @ RT for 10 h	953 ± 21	59.4 ± 2.2	3.53 ± 0.10
		@ 30 °C	(0.25 to 25 °C)	
		115 ± 30	169 ± 4.9	
		@ 60 °C	(100 to 175 °C)	
ER-B-50	Toolfusion 1A/1B 100:50 by weight cured @ RT for 5 h		53.9 ± 2.5	3.71 ± 0.17
			(0.25 to 25 °C)	
			171 ± 3.5	
			(100 to 175 °C)	

* The viscosity shown is for the epoxy resin without curing agent.

**Table 2 sensors-20-03230-t002:** Summary of zones and parameters characterized for CNTY/ER-A and CNTY/ER-B.

Zone	Time Interval	Electrical Resistance Change	Description of Zone	Focus/Parameter Characterized
I	*t*_0_–*t*_1_	*R*_0_*^II^*–*R*_0_	Initial constant temperature (stabilization for 10 min).	Signal to noise ratio, Equation (1).
II	*t*_1_–*t*_2_	*R*_0_^*III*^–*R*_0_*^II^*	Ramping from *T*_0_ to *T*_1_ at a heating rate of ~1.8 °C/min.	Thermoresistive sensitivity of the CNTY under heating, Equation (2).
III	*t*_2_–*t*_3_	*R*_0_^*IV*^–*R*_0_*^III^*	Dwell at curing temperature (*T*_1_).	Effect of CNTY/resin interaction during curing, Equation (3).
IV	*t*_3_–*t*_4_	*R*_f_–*R*_0_*^IV^*	Cooling down to *T*_0_ (cooling rate of ~0.5 °C/min).	Thermoresistive sensitivity of CNTY embedded in resin and freestanding CNTY during cooling, Equation (4), and normalized residual electrical resistance, Equation (5).

**Table 3 sensors-20-03230-t003:** Characterization parameters extracted from experiments of CNTY/ER-A and CNTY/ER-B (average and one standard deviation).

Zone	Parameter	CNTY/ER-A	CNTY/ER-B
I	*SNR^CNTY^* (dB)	29.0 ± 4.5	
II	*α_H_^CNTY^* (°C^−1^)	−8.5 × 10^−4^ ± 1.0 × 10^−5^	
III	(*ΔR*/*R*_0_)*_III_* (%)	12.0 ± 1.4	9.0 ± 0.4
	(*ΔR*/*R*_0_)*_III_^CNTY^* (%)	2.2 ± 0.2	0.6 ± 0.05
	*ERP* (%) *=* (*ΔR*/*R*_0_)*_III_* − (*ΔR*/*R*_0_)*_III_^CNTY^*	9.7 ± 1.1	8.4 ± 0.4
IV	*α_C_* (°C^−1^)	−13.4 × 10^−4^ ± 3.2 × 10^−5^	−7.6 × 10^−4^ ± 3.3 × 10^−5^
	*α_C_^CNTY^* (°C^−1^)	−8.4 × 10^−4^ ± 3.2 × 10^−5^	−6.2 × 10^−4^ ± 2.5 × 10^−5^
		(25 °C ≤ *T* ≤ 130 °C)	(25 °C ≤ *T* ≤ 60 °C)
-	(*ΔR*/*R*_0_)*_RES_* (%)	9.0 ± 4.1	−9.0 ± 2.9

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
