# Peer review of "Electrical Resistance Sensing of Epoxy Curing Using an Embedded Carbon Nanotube Yarn"

_sensors, 2020, doi:10.3390/s20113230_

Round 1

Reviewer 1 Report

Functional epoxy/nano-carbon composites are a field of intensive research in recent years. This manuscript describes electrical behavior of epoxy composites with carbon nanotube yarns. The introduction is correct, as well as results are interesting and convincingly explained. In my opinion, this contribution may be published in Sensors. Minor comments are given below.

  • Epikure is a trade name of a group of curing agents. Please, give a number of used EPIKURE curing agent
  • Analogically, please, provide a number of Toolfusion resin system
  • It should be mentioned, that also other effects, which were not analyzed in this work, might influence the electrical properties of the samples, eg. chemical formulas of resin components, as it was proven elsewhere [39a]

[39a] S. Kugler, K. Kowalczyk, T. Spychaj, Influence of synthetic and bio-based amine curing agents on properties of solventless epoxy varnishes and coatings with carbon nanofillers, Prog. Org. Coat. 109 (2017) 83-91, doi: 10.1016/j.porgcoat.2017.04.033

Reviewer 2 Report

To investigate the curing effects and development of residual stresses during epoxy resin curing, through the electrical response of a single CNTY embedded in two epoxy resins with different viscosities and curing temperatures. The electrical resistance of the embedded CNTYs in different resins were tested and analyzed. This work is meaningful to explore the curing effects and development of residual stresses during epoxy resin curing. However, there are still some questions should be answered first.

1.In this work, the specific performance, especially the strain sensitivity and characterizations of the CNTY are ignored. How about the strain sensing performance of the CNTY used in this work?

2. CNTY is a very sensitive fiber under stretching, compressing or in different temperature conditions, when embedded the CNTY into different resins, during the curing processes, the ambient temperature changing, deformations caused by residual stresses may both affect the electrical resistance changing. Therefore, please explain how to clarify the specific changing during epoxy resin curing.

3.Different with carbon fiber, CNTY is a porous CNT assymbly with very high porosity (~90%) as reported in reference (Bioinspired Superelastic Electroconductive Fiber for Wearable Electronics, ACS applied materials & interfaces 2019, 11 (47), 44735-44741). In this study, the epoxy resins with different viscosities may cause different penetration depth into CNTY.  When a CNT yarn embedded in polymer, the penetration depth of the polymer may also affect its electrical performance. Whether this factor was considered by the authors?

4.The aim of this work is presented as “investigate the curing effects and development of residual stresses during epoxy resin curing”. However, the paper is not well presented, could be modified to more readable.

Reviewer 3 Report

Reviewer comments:

This paper discuss the monitoring of curing mechanism and various kind of others phenomenon’s happening during curing process using CNT yarn. This work also report the effect of curing temperature and curative content. The paper reads well, the authors have conducted a thorough and systematic characterization to understand the underlying mechanisms responsible for the resistance change. The paper is recommended for publication with minor revision. I do believe that if the followings suggestions are considered, the overall look of the manuscript could be enhanced.

  1. I understand that authors obtained this Yarns from xx company, but would be good to provide basic information on individual CNT, like dia., length etc. A high magnification TEM of CNT should work.
  2. Please correct, in materials section ER-B cured at 60 for 1.5 hr, but in table mentioned 4 hr.
  3. Please provide conductivity level of carbon black paste you used for making connections. Any comments on the stability of connections with CNT yarn and its effect on resistance values. How one can make it sure that resistance change is purely because of curing and related phenomenon’s and not because of connections integrity during curing.
  4. A one demonstrative example of using these yarns in GFRP/CFRP composites would enhance the quality of paper.
  5. A single CNT yarn sensing information could be more localized to interface, how about the bulk epoxy.

Reviewer 4 Report

This paper reports the utillisation of a CNT yarn for the sensing of resin curing via the electrical resistance change method (fractional resistance change method - ∆R/R0%). Two epoxy resins with different viscosities and curing temperatures were demonstrated varying the curing agent concentration. The paper in general is very interesting, well written and structured, however, several things need to be considered before being accepted for publication.

  1. The authors should take into account and consider previous studies where the resistance change method of a resistive reinforcement element was utillised to study the epoxy resin curing monitoring process; as well as compare the magnitude of resistance change, the trend of resistance change over the curing process, etc. (RSC Adv., 2016,6, 55514-55525)
  2. The authors should show experiments (same duration as for the curing process) about resistance change over time by applying only the current at the two external points/ electrodes as shown in figure one. Is it possible that due to joule heating, the internal resistance of the CNT yarn increases so as the voltage potential drop measured at the two internal points/ electrodes measured is varied over time.
  3. The authors should show the resistance change over time (same duration as for the curing process) applying the same temperatures as for the curing process; however without the addition of the epoxy resin/-s. This will make clear if the resistance change is affected by the different temperatures applied since the CNTs are semi-conductors as well an metals in nature that in both cases there is a dependency with temperature (normally a mixture of armchair and zig-zag/ chiral CNTs from the growth process)

The paper contains valuable information and several results to point out the importance of a CNT yarn as a functional reinforcement and sensor for the resin curing monitoring. However, several things need further discussion as well as improvements in terms of representation of the results, the experimental methods etc.

In terms of originality, scientific quality, relevance & contribution to the field and presentation, this is a manuscript of good level. The findings of the paper are sufficiently novel to warrant its publication, however, after including and considering all the major changes proposed.

Round 2

Reviewer 4 Report

The manuscript has been significantly modified and can be now accepted for publication in its current form.